# Synthesis of 1,2,3-Triazolium Ionic Liquid-Supported Chiral Imidazolidinones and Their Application in Asymmetric Alkylation Reaction

**DOI:** 10.3390/molecules24183349

**Published:** 2019-09-14

**Authors:** Yunkyung Jeong, Yunjeong Park, Jae-Sang Ryu

**Affiliations:** College of Pharmacy & Graduate School of Pharmaceutical Sciences, Ewha Womans University, 52 Ewhayeodae-gil, Seodaemun-Gu, Seoul 03760, Korea

**Keywords:** ionic liquid, 1,2,3-triazolium, chiral imidazolidinone, asymmetric alkylation, chiral auxiliaries

## Abstract

New 1,2,3-triazolium ionic liquid-supported chiral imidazolidinones were developed. The feasibility of the ionic liquid-supported imidazolidinones as chiral auxiliaries was demonstrated in sequential propionylation-alkylation-cleavage reactions, which provided the chiral product with good to excellent chemical yields (up to 90%) and high selectivities (up to 94% ee). The progress of the reactions could be monitored by TLC and NMR, and the ionic liquid-supported chiral auxiliaries could be recovered by simple extraction.

## 1. Introduction

Since the Evans’ group first developed enantioselective alkylation reactions using chiral oxazolidinone enolates in 1980 [1], oxazolidinone auxiliaries have been extensively studied and widely applied to many asymmetric reactions, such as aldol [2,3], alkylation [4,5], Diels-alder [6], conjugated addition [7], and other reactions. As shown over the course of several decades, the use of chiral oxazolidinones offers several benefits in asymmetric reactions: Easy preparation, easy installation, easy removal, high selectivity, and recyclability. However, Evans’ chiral auxiliaries are expensive and recovering them is tedious work. Due to these issues, continued efforts have been made over the past decades to develop polymer-supported forms [8] of Evans’ chiral auxiliaries as convenient and practical tools for producing chiral compound libraries. The major advantages of this method are rapid isolation of the desired chiral product and facile recovery of the polymer-supported auxiliary by simple filtration. Nevertheless, their use in asymmetric reactions has been problematic [9,10,11,12]. Indeed, a relatively narrow range of reaction conditions and substrates have been covered, and inconsistent yields and stereoselectivity levels have been observed compared to the analogous solution-phase reactions [9,10,11,12]. These problems can be ascribed to the presence of polymer supports that are insoluble in the reaction media [13,14]. In general, the heterogeneous reaction conditions of solid-phase synthesis cause several issues, including site–site interactions [15], nonlinear kinetic behavior, diffusion-limited reactivity, unequal distribution of reagents, reduced reactivity, extended reaction times, and reagent leaching. Owing to these drawbacks, soluble matrices such as poly(ethylene glycol) [16,17,18], linear polystyrenes [19], and polyethylenes [20] have drawn increasing attention as alternative supports in combinatorial synthesis [21]. However, soluble polymer-supported synthesis can be wasteful because a large amount of solvents is required to precipitate the polymer support at the end of the reaction. While many attempts to develop efficient matrices for chiral auxiliaries have been made, the results have been unsatisfactory. Currently, it is highly desirable and necessary to develop new soluble matrix-supported reusable chiral auxiliaries.

In contrast to insoluble polymers, ionic liquid-supported compounds are soluble in certain organic solvents, and are compatible with most common reagents [22]. In particular, the recovery of ionic liquid-supported material is easily accomplished by simple extraction after a reaction is complete [23]. In this context, ionic liquids have been considered as alternative soluble supports for Evans’ chiral auxiliaries. Among several potential ionic liquids, 1,2,3-triazolium ionic liquid was selected as the best ionic liquid support. Previously, we reported the use of 1,3-dialkyl-1,2,3-triazolium ionic liquids as stable and recyclable solvents [24]. The applicability of these ionic liquids as new reaction media was also shown for the Baylis-Hillman reaction, where 1,3-dialkyl-1,2,3-triazolium ionic liquids exhibited properties comparable to those of imidazolium ionic liquids in terms of thermal stability and miscibility. Compared with imidazolium ionic liquids, 1,2,3-triazolium ionic liquids are chemically inert under acidic [25] or basic [24] conditions, which make them more suitable media or supports for acid- or base-mediated reactions. Therefore, 1,2,3-triazolium ionic liquids are considered to be the best supports for Evans’ chiral auxiliaries. Herein, we first reports ionic liquid-supported Evans’ chiral auxiliaries (Figure 1). Compared to solid-supported chiral auxiliaries, 1,2,3-triazolium ionic liquid-supported chiral auxiliaries possess several advantages: i) The reaction can be conducted under homogeneous conditions; ii) it is easy to monitor progress of the reaction using TLC; iii) it is not necessary to translate solution-phase reactions to solid-phase reactions because there is no insoluble polymer support that can have a significant effect on both reactivity and selectivity; iv) ionic liquid-supported intermediates/products can be monitored by NMR after removal of the solvents; v) the product can easily be isolated and ionic liquid-supported chiral auxiliaries can be recycled by simple extraction.

## 2. Results and Discussion

### 2.1. Synthesis of Ionic Liquid-supported Imidazolidinone ***1a**–**c***

1,2,3-Triazolium-tethered imidazolidinone **1a**–**c** and **2a**–**c** were easily produced by Cu-catalyzed Huisgen 1,3-dipolar cycloaddition [26] using *n*-butyl azide and alkyne (Scheme 1), which proved useful for the synthesis of substituted 1,2,3-triazolium ionic liquids. Synthesis commenced with the azidation of commercially available *n*-butyl bromide (**3**). In situ generated *n*-butyl azide (**4**) was captured by *N*-(5-hexynyl)phthalimide via a Cu-catalyzed click reaction, which gave 1,4-disubstituted 1,2,3-triazole **5** at an 89% yield in two steps. The phthalimide moiety was then ruptured by aminolysis using hydrazine-hydrate to afford a primary amine **6** at a yield of 98%, which was readily condensed with Boc-protected amino acids in the presence of EDCI·HCl and HOBt. In this step, three amino acids carrying benzyl, *t*-butyl, and *i*-propyl substituents were coupled to **6** to synthesize benzyl-, *t*-butyl-, and *i*-propyl-substituted imidazolidinones. The subsequent treatment of the Boc-protected amines **7a**–**c** with TFA led to the production of the primary amines **8a**–**c** in almost quantitative yields. The amide moieties of **8a**–**c** were reduced with LiAlH_4_ to yield the diamines **9a**–**c**, which were converted into the imidazolidinones **10a**–**c** using CDI and Et_3_N.

At this stage, we needed to confirm that no epimerization occurred throughout the synthesis route shown in Scheme 1. Therefore, the enantiopurity of the synthesized 1,2,3-triazole-tethered (*S*)-4-benzyl-2-imidazolidinone **10a** was examined by Mosher’s method [27,28]. For this purpose, 1,2,3-triazole-tethered (*R*)-4-benzyl-2-imidazolidinone **10a′** was also prepared by the same synthetic route using Boc-d-Phe-OH instead of Boc-l-Phe-OH (Appendix A). With both 1,2,3-triazole-tethered (*S*)-4-benzyl-2-imidazolidinone **10a** and (*R*)-4-benzyl-2-imidazolidinone **10a’** in hand, the enantiopurity was determined by NMR. Synthesized 1,2,3-triazole-tethered (*S*)-4-benzyl-2-imidazolidinone **10a** and (*R*)-4-benzyl-2-imidazolidinone **10a′** were separately treated with (*R*)-Mosher’s acid chloride to obtain crude Mosher’s imide **11a** and **11a′**, respectively, as shown in Scheme 2. After analyzing the NMR spectra of the crude mixtures, it was confirmed that both **10a** and **10a′** were enantiomerically pure, and no epimerization occurred during synthesis.

The 1-butyl-3-methyl-1,2,3-triazolium ionic liquid-supported imidazolidinones **1a**–**c** were synthesized according to Scheme 3. The 1-butyl-1,2,3-triazole-tethered imidazolidinones **10a**–**c** were methylated with methyl iodide under neat conditions, which afforded 1-butyl-3-methyl-1,2,3-triazolium iodide salts **12a**–**c** in the form of sticky oils. Then, the 1-butyl-3-methyl-1,2,3-triazolium iodide salts **12a**–**c** were treated with LiNTf_2_ in deionized water to produce 1,2,3-triazolium NTf_2_ salts **1a**–**c**.

All synthesized 1-butyl-3-methyl-1,2,3-triazolium ionic liquid-supported imidazolidinones **12a**–**c** and **1a**–**c** were liquids at room temperature and considered room temperature ionic liquids (RTILs) [29]. Especially, the 1,2,3-triazolium NTf_2_ salts **1a**–**c** were liquids even at −78 °C. The 1,2,3-triazolium NTf_2_ salts **1a**–**c** dissolved well in THF at −78 °C. Even though *n*-BuLi was added in the THF solution at −78 °C, it remained as a clear solution. Thus, the NTf_2_ salts **1a**–**c** were considered useful for low-temperature asymmetric reactions.

### 2.2. Application to Asymmetric Reactions

To assess the feasibility of asymmetric alkylation reactions using the ionic liquid-supported chiral auxiliaries **1a**–**c**, sequential propionylation-alkylation-cleavage reactions beginning with **1a**–**c** (Scheme 4) began to be investigated. Propionylation of **1a**–**c** was performed using propionyl chloride and *n*-BuLi as a base at −78 °C, and yielded **13a**–**c**. All reagents were soluble in THF at −78 °C and the reaction mixture remained homogeneous throughout the reaction. The progress of the reactions was monitored by TLC and the formation of **13a**–**c** was successfully verified by ^1^H-NMR of the crude mixture after completion of the reaction. Although **13a**–**c** could be purified by column chromatography, this study demonstrated the feasibility of the ionic liquids as liquid supports for Evans’ auxiliaries without column chromatography. Therefore, after the propionylation reaction, asymmetric alkylation reactions of **13a**–**c** using 3-chlorobenzyl bromide and NaHMDS followed the propionylation immediately after the separation of the crude products **13a**–**c** from excess reagents by simple extraction. Then, the resulting crude alkylated products **14a**–**c** were treated with aqueous 2 *N* NaOH solution in 1,4-dioxane at room temperature to cleave chiral products, which gave chiral (−)-3-(3-chlorophenyl)-2-methylpropionic acid (**15**) and regenerated the crude 1,2,3-triazolium NTf_2_ salts **1a**–**c**. Upon completion of the reaction, the product was separated by simple extraction with EtOAc after acidification with 10% HCl. The isolated **15** was purified by column chromatography and the enantiomeric excess (ee) was determined by chiral HPLC after modification to the corresponding *N*-phenylpropanamide.

Table 1 shows the yields and ee’s of the sequential three-step reactions using the 1-butyl-3-methyl-1,2,3-triazolium ionic liquid-supported imidazolidinones **1a**–**c**, which functioned as efficient chiral auxiliaries in terms of selectivity. In the sequential reactions using **1a**–**c**, the ee values were as high as 91–93%, but moderate yields were observed. Therefore, each step of the sequential reactions was scrutinized. Interestingly, an *N*-methylated byproduct **13c’** was isolated in the propionylation reaction using **1c** after column chromatography (Scheme 5), the structure of which was unambiguously identified as an *N*-methylated imidazolidinone-ionic liquid **13c′** by 2D NMR experiments (NOESY, COSY, and HMBC, Appendix A). Possibly, when propionylation of **1a**–**c** was performed using *n*-BuLi solution as a base at −78 °C, the anion generated in situ by deprotonation of imidazolidinone abstracted the methyl group from the 1-butyl-3-methyl-1,2,3-triazolium cation intermolecularly. It was found that the methyl group of 1-butyl-3-methyl-1,2,3-triazolium could act as a methylating agent. Therefore, the ionic liquids were modified to improve stability.

### 2.3. Synthesis of Ionic Liquid-Supported Imidazolidinone ***2a**–**c***

As the methyl group of 1-butyl-3-methyl-1,2,3-triazolium-supported isopropyl imidazolidinone **1c** was turned out labile under the propionylation reaction conditions, it was decided to incorporate a more sterically hindered *n*-butyl group into the 1,2,3-triazolium core instead of the methyl group in order to develop more versatile supports and to avoid undesirable side reactions. Based on this revised design, the 1,3-dibutyl-1,2,3-triazolium NTf_2_ salts **2a**–**c** were prepared (Scheme 6). The disubstituted triazoles **10a**–**c** were butylated using *n*-butyl iodide under neat conditions, which afforded 1,3-dibutyl-1,2,3-triazolium iodide salts **16a**–**c** in 77–92% yields. Then, the iodide salts **16a**–**c** were treated with LiNTf_2_ in deionized water for anion metathesis, which produced the 1,2,3-triazolium NTf_2_ salts **2a**–**c**. All synthesized 1,3-dibutyl-1,2,3-triazolium-supported imidazolidinones **16a**–**c** and **2a**–**c** were liquids at room temperature and were considered room temperature ionic liquids (RTILs). The 1,2,3-triazolium NTf_2_ salts **2a**–**c** were liquid even at −78 °C and were suitable for low-temperature asymmetric reactions.

### 2.4. Application to Asymmetric Reactions

With the more sterically hindered 1,3-dibutyl-1,2,3-triazolium-supported imidazolidinones **2a**–**c** in hand, the same feasibility tests as those done for ionic liquid-supported chiral auxiliaries were performed (Scheme 7). Sequential propionylation- alkylation-cleavage reactions were carried out as described above, and (−)-3-(3-chlorophenyl)-2-methylpropionic acid (**15**) was obtained. As shown in Table 2, high yields and ee’s were observed for all reactions using the 1,3-dibutyl-1,2,3-triazolium NTf_2_ salts **2a**–**c**. The crude product (−)-**15** obtained after cleavage was very pure, as shown in Figure 2. Notably, no disruption of either the imidazolidinone ring or the 1,2,3-triazolium was observed for all recovered **2a**–**c** after cleavage with 2 *N* NaOH [30,31]. In particular, the corresponding *N*-methylated byproducts were not observed and most of the recovered ionic liquid-supported chiral auxiliaries remained intact. As the reactions were conducted under homogeneous conditions, it was not necessary to translate solution-phase reactions to solid-phase reactions. Thus, the progress of the reactions could be monitored by TLC and NMR, which revealed consistent yield and selectivity. The inconsistent yield and selectivity problems that originated from heterogeneous solid-phase reactions were resolved by the use of ionic liquid-supported imidazolidinones.

It is particularly interesting whether the presence of the ionic liquid-tether lowers the ee or not. Thus, the enantiomeric excesses of the reactions using **2a** with those using Evans’ auxiliary (Scheme 8) were directly compared. The reactions were conducted under the standard conditions used in the Scheme 7, and observed 91% ee for (−)-**15**. As expected, the obtained ee’s for both reactions were comparable (91% ee for Scheme 8 versus 90% ee for Table 2, entry 1), and no decrease of the ee’s was observed. The 1,2,3-triazolium ionic liquid-tether did not affect the stereoselectivity**.**

## 3. Materials and Methods

### 3.1. General Methods

All reactions were performed in oven-dried glassware fitted with a glass stopper under positive pressure of Ar with magnetic stirring, unless otherwise noted. The air- and moisture-sensitive liquids and solutions were transferred via syringe or stainless-steel cannula. TLC was performed on 0.25 mm E. Merck silica gel 60 F254 plates (Merck, Kenilworth, IL, USA) and visualized under UV light (254 nm) or by staining with cerium ammonium molybdenate (CAM), potassium permanganate (KMnO_4_) or *p*-anisaldehyde. Flash chromatography was performed on E. Merck 230–400 mesh silica gel 60 (Merck). The reagents were purchased from commercial suppliers, and used without further purification unless otherwise noted. The solvents were distilled from proper drying agents (CaH_2_ or Na wire) under Ar atmosphere at 760 mm Hg. All moisture- and/or oxygen-sensitive solids were handled and stored in a glove box under N_2_. The NMR spectra were recorded on at 24 °C. The chemical shifts were expressed in ppm relative to TMS (^1^H, 0 ppm), CDCl_3_ (^1^H, 7.26 ppm; ^13^C, 77.2 ppm), DMSO-*d*_6_ (^1^H, 2.50 ppm; ^13^C, 39.5 ppm), acetone-*d*_6_ (^1^H, 2.05 ppm; ^13^C, 206.3, 29.9 ppm), CD_3_OD (^1^H, 3.31 ppm; ^13^C, 49.1 ppm) or C_6_H_5_F (^19^F, −113.15 ppm); coupling constants were expressed in Hz. The high resolution mass spectra (HRMS) were obtained by ESI (TOF analyzer) or FAB (magnetic sector analyzer). The optical rotations were determined in acetone, CHCl_3_ or MeOH. Chiral HPLC analysis for the determination of the enantomeric excess (ee) was performed on a chiral Lux 5u Cellulose-1 column (5 μm, 1000 Å, 250 × 4.6 mm). The solvents were eluted at a flow rate of 0.8 mL/min at room temperature using a binary solvent system (solvent A: hexane, solvent B: isopropanol, 90% A over 30 min) with UV detection at 254 nm.

### 3.2. Synthesis

*N-[4-(1-Butyl-**1H-**1,2,3-triazol-4-yl)butyl]phthalimide (***5***).* In a screw-cap vial, a mixture of 1-bromobutane (**3**) (800 µL, 7.29 mmol) and sodium azide (949 mg, 14.6 mmol) in anhydrous DMF (11 mL) was stirred at 120 °C for 14 h. After the mixture was cooled to rt, N-(5-hexynyl)phthalimide (1.66 g, 7.29 mmol) and CuI (139 mg, 730 μmol) were added, and the solution was stirred at 80 °C for 19 h. Upon completion of the reaction, the reaction mixture was cooled to rt and concentrated in vacuo. The purification by column chromatography on silica gel (3:1 hexanes/EtOAc) afforded a desired triazole 5 (2.13 g, 89%) as a light-yellow solid. TLC: Rf 0.21 (3:1 hexanes/EtOAc). mp: 73.4–74.8 °C. 1H NMR (400 MHz, CDCl3): δ 7.84 (dd, J = 5.6, 2.8 Hz, 2H), 7.71 (dd, J = 5.6, 2.8 Hz, 2H), 7.28 (s, 1H), 4.31 (t, J = 7.2 Hz, 2H), 3.72 (t, J = 6.8 Hz, 2H), 2.77 (t, J = 6.8 Hz, 2H), 1.87 (quintet, J = 7.2 Hz, 2H), 1.77–1.71 (m, 4H), 1.35 (sextet, J = 7.2 Hz, 2H), 0.95 (t, J = 7.2 Hz, 3H). 13C NMR (100 MHz, CDCl3): δ 168.6, 147.6, 134.1, 132.3, 123.3, 120.7, 50.0, 37.8, 32.5, 28.3, 26.9, 25.3, 19.9, 13.6. LRMS (ESI) *m/z* (rel int): (pos) 327 ([M + H]^+^, 100), 282 ([M – C_3_H_8_]^+^, 100). HRMS (ESI) m/z calcd for C18H23N4O2^+^ ([M + H]^+^) 327.1816, found 327.1822.

*4-(1-Butyl-**1H-**1,2,3-triazol-4-yl)butylamine (***6***).* In a 100 mL roundbottom flask, phthalimide 5 (1.10 g, 3.38 mmol) was dissolved in anhydrous THF (5 mL) at rt. Hydrazine hydrate (1.41 mL, 28.9 mmol) was added and the mixture was stirred at 40 °C for 6 h. Upon completion of the reaction, the reaction mixture was diluted with Et_2_O (70 mL) and insoluble solid was filtered off. The filtrate was washed with aqueous 10% NaOH (3 mL) (pH 12), and dried over anhydrous Na2SO4, filtered, and concentrated in vacuo to yield analytically pure amine 6 as a colorless liquid (648 mg, 98%). TLC: Rf 0.02 (1:1 hexanes/EtOAc). ^1^H NMR (400 MHz, CDCl_3_): δ 7.26 (s, 1H), 4.31 (t, *J* = 7.2 Hz, 2H), 2.73 (t, *J* = 7.6 Hz, 2H), 2.72 (t, *J* = 7.2 Hz, 2H), 1.87 (quintet, *J* = 7.2 Hz, 2H), 1.72 (quintet, *J* = 7.6 Hz, 2H), 1.52 (m, 2H), 1.35 (sextet, *J* = 7.2 Hz, 2H), 0.95 (t, *J* = 7.2 Hz, 3H). 13C NMR (100 MHz, CDCl^3^): δ 147.9, 120.4, 49.8, 41.9, 33.3, 32.3, 26.7, 25.5, 19.7, 13.4. LRMS (ESI) *m/z* (rel int): (pos) 219 ([M + Na]^+^, 100), 180 ([M − NH_2_]^+^, 100). HRMS (ESI) m/z calcd for C10H20N4Na^+^ ([M + Na]^+^) 219.1580, found 219.1578.

*(S)-tert-Butyl [1-((4-(1-butyl-1H-1,2,3-triazol-4-yl)butyl)amino)-1-oxo-3-phenylpropan-2-yl]carbamate (***7a***).* In a 50 mL roundbottom flask, amine 6 (261 mg, 1.33 mmol), Boc-l-Phe-OH (353 mg, 1.33 mmol), EDCI·HCl (286 mg, 1.46 mmol), and HOBt·hydrate (198 mg, 1.46 mmol) were dissolved in anhydrous CH_2_Cl_2_ (5.8 mL) at rt. The mixture was stirred at rt for 5 h. Upon completion of the reaction, 10% wt citric acid (11 mL) was added and the two layers were separated. The aqueous layer was extracted with CH_2_Cl_2_ (2 × 30 mL) and combined organic extracts were washed with saturated aqueous NaHCO_3_ (15 mL) and brine (15 mL) sequentially. It was then dried over anhydrous MgSO_4_, filtered, and concentrated in vacuo. The purification by column chromatography on silica gel (20:1 CH_2_Cl_2_/MeOH) afforded a desired amide 7a as a white foam (566 mg, 96%). TLC: R*_f_* 0.19 (1:2 hexanes/EtOAc). mp: 98.8–100.8 °C. $[\alpha {]}_{D}^{21.4}$ = +4.2 (c 1.0, CHCl_3_). ^1^H NMR (400 MHz, CDCl_3_): δ 7.30–7.19 (m, 6H), 5.82 (br s, 1H), 5.05 (br s, 1H), 4.31 (t, *J* = 7.2 Hz, 2H), 4.27 (q, *J* = 7.6 Hz, 1H), 3.19 (q, *J* = 6.4 Hz, 2H), 3.07 (dd, *J* = 13.6, 6.4 Hz, 1H), 3.02 (dd, *J* = 13.6, 7.6 Hz, 1H), 2.67 (t, *J* = 7.6 Hz, 2H), 1.88 (quintet, *J* = 7.6 Hz, 2H), 1.58 (quintet, *J* = 7.2 Hz, 2H), 1.45 (m, 2H), 1.40 (s, 9H), 1.36 (sextet, *J* = 7.2 Hz, 2H), 0.96 (t, *J* = 7.2 Hz, 3H). ^13^C NMR (100 MHz, CDCl_3_): δ 171.3, 155.6, 147.7, 137.0, 129.5, 128.8, 127.1, 120.7, 80.3, 56.2, 50.1, 39.3, 39.0, 32.5, 28.9, 28.4, 26.7, 25.2, 19.9, 13.7. LRMS (ESI) *m*/*z* (rel int): (pos) 444 ([M + H]^+^, 100), 388 ([M − C_4_H_7_]^+^, 75). HRMS (ESI) m/z calcd for C24H38N5O3+ ([M + H]^+^) 444.2969, found 444.2977.

*(R)-tert-Butyl [1-((4-(1-butyl-1H-1,2,3-triazol-4-yl)butyl)amino)-1-oxo-3-phenylpropan-2-yl]carbamate (***7a′***).***7a′** was synthesized according to the procedure described for the synthesis of **7a**. Amine 6 (719 mg, 3.66 mmol), Boc-d-Phe-OH (991 mg, 3.66 mmol), EDCI·HCl (772 mg, 4.03 mmol), and HOBt·hydrate (544 mg, 4.03 mmol) were used. Reaction time: 6 h. Yield: 1.60 g, 98%. White solid. TLC: R*_f_* 0.19 (1:2 hexanes/EtOAc). mp: 107.1–109.1 °C. ${\left[\alpha \right]}_{D}^{22.5}$ = −5.7 (c 1.0, CHCl_3_). ^1^H NMR (400 MHz, CDCl_3_): δ 7.30–7.19 (m, 6H), 5.89 (br s, 1H), 5.08 (br s, 1H), 4.31 (t, *J* = 7.2 Hz, 2H), 4.28 (q, *J* = 7.6 Hz, 1H), 3.19 (q, *J* = 6.8 Hz, 2H), 3.07 (dd, *J* = 13.6, 6.4 Hz, 1H), 3.02 (dd, *J* = 13.6, 7.6 Hz, 1H), 2.67 (t, *J* = 7.2 Hz, 2H), 1.87 (quintet, *J* = 7.2 Hz, 2H), 1.59 (quintet, *J* = 7.6 Hz, 2H), 1.44 (m, 2H), 1.40 (s, 9H), 1.36 (sextet, *J* = 7.2 Hz, 2H), 0.96 (t, *J* = 7.2 Hz, 3H). ^13^C NMR (100 MHz, CDCl_3_): δ 171.0, 155.4, 147.5, 136.9, 129.3, 128.6, 126.9, 120.5, 80.1, 56.1, 49.9, 39.1, 38.8, 32.3, 28.8, 28.3, 26.5, 25.1, 19.7, 13.5. LRMS (ESI) *m*/*z* (rel int): (pos) 444 ([M + H]^+^, 100), 388 ([M − C_4_H_7_]^+^, 23). HRMS (ESI) m/z calcd for C24H38N5O3^+^ ([M + H]^+^) 444.2969, found 444.2969.

*(S)-tert-Butyl [1-((4-(1-butyl-1H-1,2,3-triazol-4-yl)butyl)amino)-3,3-dimethyl-1-oxobutan-2-yl]carbamate (***7b***).***7b** was synthesized according to the procedure described for the synthesis of **7a**. Amine 6 (813 mg, 4.14 mmol), Boc-*tert*-Leu-OH (931 mg, 3.95 mmol), EDCI·HCl (832 mg, 4.34 mmol), and HOBt·hydrate (587 mg, 4.34 mmol) were used. Reaction time: 14 h. Yield: 1.52 g, 94%. Colorless oil. TLC: R*_f_* 0.43 (10:1 CH_2_Cl_2_/MeOH). ${\left[\alpha \right]}_{D}^{22.7}$ = +1.3 (c 1.0, CHCl_3_). ^1^H NMR (400 MHz, CDCl_3_): δ 7.27 (s, 1H), 5.78 (br s, 1H), 5.25 (br d, *J* = 8.8 Hz, 1H), 4.31 (t, *J* = 7.2 Hz, 2H), 3.75 (br d, *J* = 9.2 Hz, 1H), 3.29 (m, 2H), 2.73 (t, *J* = 7.2 Hz, 2H), 1.87 (quintet, *J* = 7.2 Hz, 2H), 1.72 (quintet, *J* = 7.2 Hz, 2H), 1.59 (m, 2H), 1.43 (s, 9H), 1.35 (sextet, *J* = 7.2 Hz, 2H), 0.98 (s, 9H), 0.96 (t, *J* = 7.2 Hz, 3H). ^13^C NMR (100 MHz, CDCl_3_): δ 170.9, 155.8, 147.5, 120.6, 79.6, 62.5, 49.9, 39.1, 34.5, 32.3, 28.9, 28.3, 26.7, 26.6, 25.1, 19.7, 13.5. LRMS (ESI) *m/z* (rel int): (pos) 410 ([M + H]^+^, 100), 354 ([M – C_4_H_7_]^+^, 22). HRMS (ESI) m/z calcd for C21H40N5O3^+^ ([M + H]^+^) 410.3126, found 410.3129.

*(S)-tert-Butyl [1-((4-(1-butyl-1H-1,2,3-triazol-4-yl)butyl)amino)-3-methyl-1-oxobutan-2-yl]carbamate (***7c***).***7c** was synthesized according to the procedure described for the synthesis of **7a**. Amine 6 (690 mg, 3.51 mmol), Boc-l-Val-OH (727 mg, 3.35 mmol), EDCI·HCl (706 mg, 3.68 mmol), and HOBt·hydrate (498 mg, 3.68 mmol) were used. The crude mixture was purified by column chromatography on silica gel (30:1 CH_2_Cl_2_/MeOH). Reaction time: 8 h. Yield: 1.26 g, 95%. White solid. TLC: R*_f_* 0.52 (10:1 CH_2_Cl_2_/MeOH). mp: 81.5–83.5 °C. ${\left[\alpha \right]}_{D}^{21.6}$ = −15.4 (c 1.0, MeOH). ^1^H NMR (400 MHz, CDCl_3_): δ 7.27 (s, 1H), 6.03 (br s, 1H), 5.02 (br s, 1H), 4.31 (t, *J* = 7.2 Hz, 2H), 3.84 (dd, *J* = 8.8, 6.0 Hz, 1H), 3.29 (q, *J* = 6.8 Hz, 2H), 2.73 (t, *J* = 7.6 Hz, 2H), 2.14 (m, 1H), 1.87 (quintet, *J* = 7.2 Hz, 2H), 1.72 (quintet, *J* = 7.6 Hz, 2H), 1.59 (m, 2H), 1.44 (s, 9H), 1.35 (sextet, *J* = 7.2 Hz, 2H), 0.96 (t, *J* = 7.2 Hz, 3H), 0.95 (d, *J* = 6.8 Hz, 3H), 0.90 (d, *J* = 6.8 Hz, 3H). ^13^C NMR (100 MHz, CDCl_3_): δ 171.6, 155.9, 147.6, 120.6, 79.8, 60.1, 49.9, 39.1, 32.3, 30.9, 28.9, 28.3, 26.7, 25.1, 19.7, 19.3, 17.8, 13.5. LRMS (ESI) *m*/*z* (rel int): (pos) 396 ([M + H]^+^, 100), 340 ([M − C_4_H_7_]^+^, 45). HRMS (ESI) m/z calcd for C20H38N5O3^+^ ([M + H]^+^) 396.2969, found 396.2961.

*(S)-2-Amino-N-[4-(1-butyl-1H-1,2,3-triazol-4-yl)butyl]-3-phenylpropanamide (***8a***).* In a 250 mL roundbottom flask, Boc protected amine **7a** (1.87 g, 4.21 mmol) was dissolved in CH_2_Cl_2_ (33 mL) at rt under air. A solution of TFA in CH_2_Cl_2_ (20% *v*/*v*, 48 mL) was added dropwise for 2 h. Upon completion of the reaction, the reaction mixture was diluted with CH_2_Cl_2_ (30 mL), cooled to 0 °C, and basified with saturated aqueous NaHCO_3_ (800 mL). The aqueous layer was extracted with CH_2_Cl_2_ (3 × 100 mL). The combined organic extracts were then dried over anhydrous Na_2_SO_4_, filtered, and concentrated in vacuo to yield analytically pure amine 8a as a white solid (1.52 g, quantitative yield). TLC: R*_f_* 0.65 (5:1 CH_2_Cl_2_/MeOH). mp: 57.2–59.2 °C. ${\left[\alpha \right]}_{D}^{21.6}$ = −44.3 (c 1.0, CHCl_3_). ^1^H NMR (400 MHz, CDCl_3_): δ 7.33–7.21 (m, 6H), 4.31 (t, *J* = 7.2 Hz, 2H), 3.62 (dd, *J* = 9.2, 4.0 Hz, 1H), 3.31–3.25 (m, 3H), 2.73 (t, *J* = 7.2 Hz, 2H), 2.71 (dd, *J* = 13.2, 9.2 Hz, 1H), 1.87 (quintet, *J* = 7.2 Hz, 2H), 1.70 (quintet, *J* = 7.2 Hz, 2H), 1.57 (m, 2H), 1.36 (sextet, *J* = 7.2 Hz, 2H), 0.96 (t, *J* = 7.2 Hz, 3H). ^13^C NMR (100 MHz, CDCl_3_): δ 174.3, 147.8, 138.1, 129.4, 128.8, 126.9, 120.7, 56.6, 50.0, 41.2, 38.9, 32.4, 29.2, 26.9, 25.3, 19.9, 13.6. LRMS (ESI) *m*/*z* (rel int): (pos) 344 ([M + H]^+^, 100), 180 ([M − *n*-Bu − NH_2_ − Bn]^+^, 33). HRMS (ESI) m/z calcd for C19H30N5O^+^ ([M + H]^+^) 344.2445, found 344.2442.

*(R)-2-Amino-N-[4-(1-butyl-1H-1,2,3-triazol-4-yl)butyl]-3-phenylpropanamide**(***8a′***).***8a′** was synthesized according to the procedure described for the synthesis of **8a**. Amine **7a′** (818 mg, 1.84 mmol) and a solution of TFA in anhydrous CH_2_Cl_2_ (20% *v*/*v*, 16 mL) were used. Reaction time: 1 h. Yield: 615 mg, 97%. White solid. TLC: R*_f_* 0.65 (5:1 CH_2_Cl_2_/MeOH). mp: 62.5–64.5 °C. ${\left[\alpha \right]}_{D}^{20.6}$ = −16.9 (c 1.0, MeOH). ^1^H NMR (400 MHz, CDCl_3_): δ 7.33–7.21 (m, 6H), 4.32 (t, *J* = 7.2 Hz, 2H), 3.59 (dd, *J* = 9.2, 4.0 Hz, 1H), 3.29 (t, *J* = 7.2 Hz, 2H), 3.27 (dd, *J* = 13.6, 4.0 Hz, 1H), 2.73 (t, *J* = 7.2 Hz, 2H), 2.68 (dd, *J* = 13.6, 9.2 Hz, 1H), 1.87 (quintet, *J* = 7.2 Hz, 2H), 1.70 (quintet, *J* = 7.2 Hz, 2H), 1.56 (m, 2H), 1.36 (sextet, *J* = 7.2 Hz, 2H), 0.96 (t, *J* = 7.2 Hz, 3H). ^13^C NMR (100 MHz, CDCl_3_): δ 174.2, 147.7, 138.0, 129.3, 128.7, 126.8, 120.6, 56.6, 49.9, 41.1, 38.8, 32.3, 29.1, 26.8, 25.2, 19.7, 13.5. LRMS (ESI) *m*/*z* (rel int): (pos) 344 ([M + H]^+^, 100), 180 ([M − C_4_H_9_ − NH_2_ − Bn]^+^, 37). HRMS (ESI) m/z calcd for C19H30N5O^+^ ([M + H]^+^) 344.2445, found 344.2445.

*(S)-2-Amino-N-[4-(1-butyl-1H-1,2,3-triazol-4-yl)butyl]-3,3-dimethylbutanamide (***8b***)*. **8b** was synthesized according to the procedure described for the synthesis of **8a**. Amine **7**b (1.16 g, 2.84 mmol) and a solution of TFA in anhydrous CH_2_Cl_2_ (20% *v/v*, 30 mL) were used. Reaction time: 1.5 h. Yield: 871 mg, 99%. White solid. TLC: R*_f_* 0.17 (10:1 CH_2_Cl_2_/MeOH). mp: 45.0–47.0 °C. ${\left[\alpha \right]}_{D}^{21.8}$ = +27.8 (c 1.0, MeOH). ^1^H NMR (400 MHz, CDCl_3_): δ 7.27 (s, 1H), 6.81 (br s, 1H), 4.31 (t, *J* = 7.2 Hz, 2H), 3.29 (m, 2H), 3.09 (s, 1H), 2.74 (t, *J* = 7.2 Hz, 2H), 1.87 (quintet, *J* = 7.2 Hz, 2H), 1.73 (quintet, *J* = 7.2 Hz, 2H), 1.58 (m, 2H), 1.35 (sextet, *J* = 7.2 Hz, 2H), 0.99 (s, 9H), 0.95 (t, *J* = 7.2 Hz, 3H). ^13^C NMR (100 MHz, CDCl_3_): δ 173.5, 147.6, 120.6, 64.4, 49.9, 38.7, 34.0, 32.3, 29.1, 26.8, 26.7, 25.2, 19.7, 13.5. LRMS (ESI) *m*/*z* (rel int): (pos) 310 ([M + H]^+^, 100), 180 ([M − *n*-Bu − NH_2_ − *t*-Bu]^+^, 37). HRMS (ESI) m/z calcd for C16H32N5O^+^ ([M + H]^+^) 310.2601, found 310.2605.

*(S)-2-Amino-N-[4-(1-butyl-1H-1,2,3-triazol-4-yl)butyl]-3-methylbutanamide (***8c***).***8c** was synthesized according to the procedure described for the synthesis of **8a**. Amine **7**c (933 mg, 2.36 mmol) and a solution of TFA in anhydrous CH_2_Cl_2_ (20% *v*/*v*, 20 mL) were used. Reaction time: 2.5 h. Yield: 687 mg, 99%. Ivory solid. TLC: R*_f_* 0.33 (10:1 CH_2_Cl_2_/MeOH). mp: 58.9–60.9 °C. ${\left[\alpha \right]}_{D}^{21.9}$ = +5.9 (c 1.0, MeOH). ^1^H NMR (400 MHz, CDCl_3_): δ 7.36 (br s, 1H), 7.27 (s, 1H), 4.31 (t, *J* = 7.2 Hz, 2H), 3.31 (m, 2H), 3.23 (d, *J* = 3.6 Hz, 1H), 2.74 (t, *J* = 7.6 Hz, 2H), 2.30 (m, 1H), 1.87 (quintet, *J* = 7.2 Hz, 2H), 1.73 (quintet, *J* = 7.6 Hz, 2H), 1.58 (m, 2H), 1.35 (sextet, *J* = 7.2 Hz, 2H), 0.98 (d, *J* = 6.8 Hz, 3H), 0.95 (t, *J* = 7.2 Hz, 3H), 0.81 (d, *J* = 6.8 Hz, 3H). ^13^C NMR (100 MHz, CDCl_3_): δ 174.2, 147.7, 120.6, 60.2, 49.9, 38.7, 32.3, 30.8, 29.2, 26.8, 25.2, 19.8, 19.7, 16.1, 13.5. LRMS (ESI) *m*/*z* (rel int): (pos) 296 ([M + H]^+^, 100), 180 ([M − *n*-Bu − NH_2_ − *i*-Pr]^+^, 53). HRMS (ESI) m/z calcd for C15H30N5O^+^ ([M + H]^+^) 296.2445, found 296.2450.

*(S)-N1-[4-(1-Butyl-1H-1,2,3-triazol-4-yl)butyl]-3-phenylpropane-1,2-diamine (***9a***).* In a 100 mL roundbottom flask, amide 8a (1.40 g, 4.09 mmol) was dissolved in anhydrous THF (16 mL) at rt. The reaction mixture was cooled to 0 °C and LiAlH_4_ (931 mg, 24.5 mmol) was added. It was then stirred at 0 °C for 30 min and heated under reflux for 7 h. Upon completion of the reaction, the resulting grey suspension was cooled to 0 °C, quenched with saturated aqueous Na_2_SO_4_, and diluted with EtOAc (70 mL). The precipitate was filtered off, and the filtrate was concentrated by rotary evaporation. The residue was diluted with CH_2_Cl_2_ (100 mL), dried over anhydrous K_2_CO_3_, filtered, and concentrated in vacuo. The purification by column chromatography on silica gel (15:1 CH_2_Cl_2_/MeOH→10:1 CH_2_Cl_2_/MeOH→MeOH) afforded a desired diamine 9a as a pale yellow solid (1.17 g, 87%). TLC: R*_f_* 0.04 (MeOH). mp: 38.1–40.1 °C. ${\left[\alpha \right]}_{D}^{21.0}$ = +6.3 (c 1.0, MeOH). ^1^H NMR (400 MHz, CDCl_3_): δ 7.32−7.18 (m, 6H), 4.31 (t, *J* = 7.2 Hz, 2H), 3.09 (ddd, *J* = 13.6, 8.8, 4.8 Hz, 1H), 2.78 (dd, *J* = 13.6, 4.8 Hz, 1H), 2.73 (t, *J* = 7.2 Hz, 2H), 2.70 (dd, *J* = 11.6, 4.8 Hz, 1H), 2.64 (q, *J* = 7.2 Hz, 2H), 2.49 (dd, *J* = 13.6, 8.8 Hz, 1H), 2.45 (dd, *J* = 11.6, 8.8 Hz, 1H), 1.87 (quintet, *J* = 7.2 Hz, 2H), 1.72 (quintet, *J* = 7.2 Hz, 2H), 1.55 (m, 2H), 1.35 (sextet, *J* = 7.2 Hz, 2H), 0.95 (t, *J* = 7.2 Hz, 3H). ^13^C NMR (100 MHz, CD_3_OD): δ 149.0, 140.2, 130.4, 129.6, 127.5, 123.1, 56.3, 53.3, 51.0, 50.5, 43.2, 33.4, 30.0, 28.3, 26.1, 20.7, 13.8. LRMS (ESI) *m*/*z* (rel int): (pos) 330 ([M + H]^+^, 100), 282 ([C_15_H_32_N_5_]^+^, 11), 180 ([M − C_2_H_6_N_2_ − Bn]^+^, 23). HRMS (ESI) m/z calcd for C19H32N5+ ([M + H]^+^) 330.2652, found 330.2659.

*(R)-N1-[4-(1-Butyl-1H-1,2,3-triazol-4-yl)butyl]-3-phenylpropane-1,2-diamine**(***9a′***).***9a′** was synthesized according to the procedure described for the synthesis of **9a**. Amide **8**a**′** (425 mg, 1.24 mmol) and LiAlH_4_ (299 mg, 7.88 mmol) were used. Reaction time: 6 h. Yield: 361 mg, 88%. White solid. TLC: R*_f_* 0.04 (MeOH). mp: 47.2–49.2 °C. ${\left[\alpha \right]}_{D}^{21.1}$ = −7.6 (c 1.0, MeOH). ^1^H NMR (400 MHz, CDCl_3_): δ 7.32–7.18 (m, 6H), 4.31 (t, *J* = 7.2 Hz, 2H), 3.09 (br s, 1H), 2.78 (dd, *J* = 13.2, 4.8 Hz, 1H), 2.73 (t, *J* = 7.6 Hz, 2H), 2.68 (m, 1H), 2.64 (q, *J* = 7.2 Hz, 2H), 2.50 (dd, *J* = 13.2, 8.8 Hz, 1H), 2.45 (dd, *J* = 11.2, 8.8 Hz, 1H), 1.87 (quintet, *J* = 7.2 Hz, 2H), 1.72 (quintet, *J* = 7.6 Hz, 2H), 1.55 (m, 2H), 1.35 (sextet, *J* = 7.2 Hz, 2H), 0.95 (t, *J* = 7.2 Hz, 3H). ^13^C NMR (100 MHz, CD_3_OD): δ 149.0, 140.2, 130.4, 129.6, 127.5, 123.2, 56.4, 53.3, 51.0, 50.5, 43.2, 33.4, 30.0, 28.3, 26.1, 20.7, 13.8. LRMS (ESI) *m*/*z* (rel int): (pos) 330 ([M + H]^+^, 100), 282 ([C_15_H_32_N_5_]^+^, 48), 180 ([M − C_2_H_6_N_2_ − Bn]^+^, 22). HRMS (ESI) m/z calcd for C19H32N5^+^ ([M + H]^+^) 330.2652, found 330.2658.

*(S)-N1-[4-(1-Butyl-1H-1,2,3-triazol-4-yl)butyl]-3,3-dimethylbutane-1,2-diamine (***9b***).***9b** was synthesized according to the procedure described for the synthesis of **9a**. Amide **8b** (2.30 g, 7.42 mmol) and LiAlH_4_ (1.69 g, 44.5 mmol) were used. Reaction time: 24 h. Yield: 1.47 g, 67% (brsm: 83%). Colorless oil. TLC: R*_f_* 0.07 (MeOH). ${\left[\alpha \right]}_{D}^{21.7}$ = +23.8 (c 1.0, MeOH). ^1^H NMR (400 MHz, CDCl_3_): δ 7.25 (s, 1H), 4.31 (t, *J* = 7.2 Hz, 2H), 2.77 (dd, *J* = 11.2, 2.4 Hz, 1H), 2.74 (t, *J* = 7.6 Hz, 2H), 2.67 (dt, *J* = 11.2, 7.2 Hz, 1H), 2.59 (dt, *J* = 11.2, 7.2 Hz, 1H), 2.46 (dd, *J* = 11.2, 2.4 Hz, 1H), 2.21 (t, *J* = 11.2 Hz, 1H), 1.87 (quintet, *J* = 7.2 Hz, 2H), 1.73 (quintet, *J* = 7.6 Hz, 2H), 1.57 (m, 2H), 1.35 (sextet, *J* = 7.2 Hz, 2H), 0.95 (t, *J* = 7.2 Hz, 3H), 0.88 (s, 9H). ^13^C NMR (100 MHz, DMSO-*d*_6_): δ 146.8, 121.5, 59.1, 51.3, 49.2, 48.8, 33.3, 31.7, 29.3, 26.9, 26.3, 25.0, 19.1, 13.3. LRMS (ESI) *m*/*z* (rel int): (pos) 296 ([M + H]^+^, 100), 282 ([C_15_H_32_N_5_]^+^, 21). HRMS (ESI) m/z calcd for C16H34N5^+^ ([M + H]^+^) 296.2809, found 296.2814.

*(S)-N1-[4-(1-Butyl-1H-1,2,3-triazol-4-yl)butyl]-3-methylbutane-1,2-diamine (***9c***).***9c** was synthesized according to the procedure described for the synthesis of **9a**. Amide **8c** (900 mg, 3.05 mmol) and LiAlH_4_ (482 mg, 18.3 mmol) were used. Reaction time: 24 h. Yield: 694 mg, 81% (brsm: 89%). Milky oil. TLC: R*_f_* 0.05 (MeOH). ${\left[\alpha \right]}_{D}^{21.8}$ = +16.3 (c 1.0, MeOH). ^1^H NMR (400 MHz, CDCl_3_): δ 7.25 (s, 1H), 4.31 (t, *J* = 7.2 Hz, 2H), 2.73 (t, *J* = 7.6 Hz, 2H), 2.69–2.56 (m, 4H), 2.34 (dd, *J* = 11.6, 9.6 Hz, 1H), 1.87 (quintet, *J* = 7.2 Hz, 2H), 1.72 (quintet, *J* = 7.6 Hz, 2H), 1.60–1.52 (m, 3H), 1.35 (sextet, *J* = 7.2 Hz, 2H), 0.95 (t, *J* = 7.2 Hz, 3H), 0.91 (d, *J* = 6.8 Hz, 3H), 0.89 (d, *J* = 6.8 Hz, 3H). ^13^C NMR (100 MHz, CDCl_3_): δ 148.1, 120.4, 56.6, 54.2, 49.9 (2C), 32.5, 32.3, 29.8, 27.3, 25.6, 19.8, 19.4, 17.8, 13.5. LRMS (ESI) *m*/*z* (rel int): (pos) 282 ([M + H]^+^, 82), 141 ([M − nBu − C_2_HN_3_ − NH_2_]^+^, 100). HRMS (ESI) m/z calcd for C15H32N5^+^ ([M + H]^+^) 282.2652, found 282.2662.

*(S)-4-Benzyl-1-[4-(1-butyl-1H-1,2,3-triazol-4-yl)butyl]imidazolidin-2-one (***10a***).* In a 100 mL roundbottom flask, diamine 9a (440 mg, 1.34 mmol) was dissolved in anhydrous DMF (1.3 mL) at rt. Et_3_N (470 µL, 3.34 mmol) and CDI (335 mg, 2.00 mmol) were added. The resulting reaction mixture was stirred at 120 °C for 13 h. Upon completion of the reaction, the reaction mixture was cooled to rt, and concentrated in vacuo. The purification by column chromatography on silica gel (15:1 CH_2_Cl_2_/MeOH) afforded a desired imidazolidinone 10a as a colorless liquid (470 mg, 99%). TLC: R*_f_* 0.49 (10:1 CH_2_Cl_2_/MeOH). ${\left[\alpha \right]}_{D}^{22.3}$ = −27.7 (c 1.0, CHCl_3_). ^1^H NMR (400 MHz, CDCl_3_): δ 7.34–7.30 (m, 2H), 7.28–7.23 (m, 2H), 7.19–7.17 (m, 2H), 4.31 (t, *J* = 7.2 Hz, 2H), 4.30 (br s, 1H), 3.86 (m, 1H), 3.50 (t, *J* = 8.8 Hz, 1H), 3.20 (m, 2H), 3.13 (dd, *J* = 8.8, 6.0 Hz, 1H), 2.83 (dd, *J* = 13.2, 6.0 Hz, 1H), 2.77 (dd, *J* = 13.2, 8.8 Hz, 1H), 2.75 (t, *J* = 7.6 Hz, 2H), 1.87 (quintet, *J* = 7.2 Hz, 2H), 1.69 (m, 2H), 1.57 (m, 2H), 1.35 (sextet, *J* = 7.2 Hz, 2H), 0.95 (t, *J* = 7.2 Hz, 3H). ^13^C NMR (100 MHz, CDCl_3_): δ 161.7, 147.8, 137.2, 129.2, 128.9, 127.0, 120.7, 51.4, 50.3, 50.0, 43.0, 42.1, 32.4, 27.2, 26.7, 25.3, 19.8, 13.6. LRMS (ESI) *m*/*z* (rel int): (pos) 356 ([M + H]^+^, 16), 282 ([C_15_H_32_N_5_]^+^, 100). HRMS (ESI) m/z calcd for C20H30N5O^+^ ([M + H]^+^) 356.2445, found 356.2440.

*(R)-4-Benzyl-1-[4-(1-butyl-1H-1,2,3-triazol-4-yl)butyl]imidazolidin-2-one (***10a′***).***10a′** was synthesized according to the procedure described for the synthesis of **10a**. Diamine **9a′** (179 mg, 542 µmol), Et_3_N (190 µL, 1.36 mmol) and CDI (132 mg, 814 µmol) were used. Reaction time: 13 h. Yield: 175 mg, 91%. Colorless liquid. TLC: R*_f_* 0.49 (10:1 CH_2_Cl_2_/MeOH). ${\left[\alpha \right]}_{D}^{22.8}$ = +26.4 (c 1.0, CHCl_3_). ^1^H NMR (400 MHz, CDCl_3_): δ 7.34–7.30 (m, 2H), 7.27–7.22 (m, 2H), 7.19–7.17 (m, 2H), 4.31 (t, *J* = 7.2 Hz, 2H), 4.29 (br s, 1H), 3.86 (m, 1H), 3.49 (t, *J* = 8.4 Hz, 1H), 3.23 (dq, *J* = 14.0, 7.2 Hz, 2H), 3.20 (m, 2H), 3.13 (dd, *J* = 8.8, 6.0 Hz, 1H), 2.82 (m, 2H), 2.75 (t, *J* = 7.6 Hz, 2H), 1.87 (quintet, *J* = 7.2 Hz, 2H), 1.69 (m, 2H), 1.56 (m, 2H), 1.35 (sextet, *J* = 7.2 Hz, 2H), 0.95 (t, *J* = 7.2 Hz, 3H). ^13^C NMR (100 MHz, CDCl_3_): δ 161.7, 147.8, 137.3, 129.2, 129.0, 127.1, 120.7, 51.5, 50.4, 50.1, 43.1, 42.2, 32.5, 27.2, 26.8, 25.4, 19.9, 13.7. LRMS (ESI) *m*/*z* (rel int): (pos) 356 ([M + H]^+^, 100), 282 ([C_15_H_32_N_5_]^+^, 95). HRMS (ESI) m/z calcd for C20H30N5O^+^ ([M + H]^+^) 356.2445, found 356.2452.

*(S)-4-(tert-Butyl)-1-[4-(1-butyl-1H-1,2,3-triazol-4-yl)butyl]imidazolidin-2-one (***10b***).***10b** was synthesized according to the procedure described for the synthesis of **10a**. Diamine **9b** (1.35 g, 4.55 mmol), Et_3_N (1.60 mL, 11.4 mmol) and CDI (1.11 g, 6.83 mmol) were used. Reaction time: 15 h. Yield: 1.46 g, quantitative yield. Colorless crystal. TLC: Rf 0.29 (30:1 CH_2_Cl_2_/MeOH (×2)). mp: 80.3–82.3 °C. ${\left[\alpha \right]}_{D}^{22.6}$ = −6.9 (c 1.0, CHCl_3_). ^1^H NMR (400 MHz, CDCl_3_): δ 7.27 (s, 1H), 4.31 (t, *J* = 7.2 Hz, 2H), 4.30 (s, 1H), 3.41–3.34 (m, 2H), 3.25 (dt, *J* = 13.6, 7.2 Hz, 1H), 3.15 (dd, *J* = 13.6, 7.2 Hz, 1H), 3.13 (m, 1H), 2.75 (t, *J* = 7.6 Hz, 2H), 1.87 (quintet, *J* = 7.2 Hz, 2H), 1.68 (m, 2H), 1.58 (m, 2H), 1.35 (sextet, *J* = 7.2 Hz, 2H), 0.95 (t, *J* = 7.2 Hz, 3H), 0.87 (s, 9H). ^13^C NMR (100 MHz, CDCl_3_): δ 161.9, 147.8, 120.6, 58.8, 49.9, 46.4, 42.9, 33.2, 32.3, 27.0, 26.6, 25.2, 25.0, 19.7, 13.5. LRMS (ESI) *m*/*z* (rel int): (pos) 322 ([M + H]^+^, 100), 282 ([C_15_H_32_N_5_]^+^, 42). HRMS (ESI) m/z calcd for C17H32N5O^+^ ([M + H]^+^) 322.2601, found 322.2603.

*(S)-4-Isopropyl-1-[4-(1-butyl-1,2,3-triazol-4-yl)butyl]imidazolidin-2-one (***10c***).***10c** was synthesized according to the procedure described for the synthesis of **10a**. Diamine **9c** (2.05 g, 7.30 mmol), Et_3_N (2.60 mL, 18.2 mmol) and CDI (1.77 g, 10.9 mmol) were used. Reaction time: 14.5 h. Yield: 1.78 g, 79%. White solid. TLC: R*_f_* 0.29 (30:1 CH_2_Cl_2_/MeOH (×2)). mp: 109.7–111.7 °C. ${\left[\alpha \right]}_{D}^{21.7}$ = −9.0 (c 1.0, MeOH). ^1^H NMR (400 MHz, CDCl_3_): δ 7.27 (s, 1H), 4.32 (s, 1H), 4.31 (t, *J* = 7.2 Hz, 2H), 3.46 (t, *J* = 8.4 Hz, 1H), 3.36 (ddd, *J* = 16.0, 7.2, 1.6 Hz, 1H), 3.26 (dt, *J* = 13.6, 7.2 Hz, 1H), 3.14 (dt, *J* = 13.6, 7.2 Hz, 1H), 3.06 (dd, *J* = 8.4, 7.2 Hz, 1H), 2.75 (t, *J* = 7.6 Hz, 2H), 1.87 (quintet, *J* = 7.2 Hz, 2H), 1.73–1.63 (m, 3H), 1.60 (m, 2H), 1.35 (sextet, *J* = 7.2 Hz, 2H), 0.95 (t, *J* = 7.2 Hz, 3H), 0.92 (d, *J* = 6.8 Hz, 3H), 0.88 (d, *J* = 6.8 Hz, 3H). ^13^C NMR (100 MHz, CDCl_3_): δ 162.0, 147.8, 120.6, 55.8, 49.9, 48.8, 43.0, 33.1, 32.3 27.1, 26.6, 25.3, 19.7, 18.4, 17.9, 13.5. LRMS (ESI) *m*/*z* (rel int): (pos) 308 ([M + H]^+^, 100), 282 ([C_15_H_32_N_5_]^+^, 25). HRMS (ESI) m/z calcd for C16H30N5O^+^ ([M + H]^+^) 308.2445, found 308.2450.

*(S)-4-[4-(4-Benzyl-2-oxoimidazolidin-1-yl)butyl]-1-butyl-3-methyl-1H-1,2,3-triazolium iodide (***12a***).* In a screw-cap vial, a mixture of **10a** (161 mg, 454 μmol) and MeI (28.3 μL, 454 μmol) was stirred at 80 °C. After 24 h, the reaction mixture was cooled to rt. The purification by column chromatography on silica gel (15:1 CH_2_Cl_2_/MeOH) afforded a desired compound **12a** (183 mg, 81%) as a light-yellow sticky gum. TLC: R*_f_* 0.15 (10:1 CH_2_Cl_2_/MeOH). ${\left[\alpha \right]}_{D}^{22.3}$ = −16.0 (c 1.0, CHCl_3_). ^1^H NMR (400 MHz, CDCl_3_): δ 9.33 (s, 1H), 7.31 (m, 2H), 7.25–7.19 (m, 3H), 4.68 (t, *J* = 7.6 Hz, 2H), 4.36 (s, 1H), 4.28 (s, 3H), 3.97 (m, 1H), 3.62 (t, *J* = 8.8 Hz, 1H), 3.32 (dt, *J* = 14.4, 7.2 Hz, 1H), 3.18 (dd, *J* = 8.8, 6.4 Hz, 1H), 3.16 (m, 1H), 3.04 (m, 2H), 2.87 (dd, *J* = 13.6, 5.6 Hz, 1H), 2.81 (dd, *J* = 13.6, 7.6 Hz, 1H), 2.03 (quintet, *J* = 7.6 Hz, 2H), 1.84 (m, 2H), 1.70 (m, 2H), 1.44 (sextet, *J* = 7.6 Hz, 2H), 0.98 (t, *J* = 7.6 Hz, 3H). ^13^C NMR (100 MHz, CDCl_3_): δ 162.1, 144.5, 137.1, 129.6, 129.3, 128.8, 126.8, 54.0, 51.3, 50.2, 42.0, 41.8, 38.9, 31.3, 26.6, 24.0, 23.2, 19.5, 13.5. LRMS (FAB) *m*/*z* (rel int): (pos) 370 ([C_21_H_32_N_5_O]^+^, 100), 278 ([C_21_H_32_N_5_O − Bn]^+^, 15). HRMS (FAB) *m*/*z* calcd for C_21_H_32_N_5_O^+^ 370.2601, found 370.2603.

*(S)-4-[4-[4-(tert-Butyl)-2-oxoimidazolidin-1-yl])butyl]-1-butyl-3-methyl-1H-1,2,3-triazolium iodide (***12b***).***12b** was synthesized according to the procedure described for the synthesis of **12a**. **10b** (1.46 g, 4.54 mmol) and MeI (290 µL, 4.54 mmol) were used. Reaction time: 24 h. Yield: 1.80 g, 86%. Yellow gum. TLC: R*_f_* 0.47 (10:1 CH_2_Cl_2_/MeOH). ${\left[\alpha \right]}_{D}^{22.8}$ = −8.1 (c 1.0, CHCl_3_). ^1^H NMR (400 MHz, CDCl_3_): δ 9.41 (s, 1H), 4.70 (t, *J* = 7.6 Hz, 2H), 4.53 (br s, 1H), 4.26 (s, 3H), 3.51–3.45 (m, 2H), 3.36 (dt, *J* = 14.4, 6.8 Hz, 1H), 3.22–3.14 (m, 2H), 3.04 (td, *J* = 8.0, 2.8 Hz, 2H), 2.04 (quintet, *J* = 7.6 Hz, 2H), 1.86 (quintet, *J* = 7.6 Hz, 2H), 1.73 (sextet, *J* = 7.6 Hz, 2H), 1.44 (sextet, *J* = 7.6 Hz, 2H), 0.99 (t, *J* = 7.6 Hz, 3H), 0.88 (s, 9H). ^13^C NMR (100 MHz, CDCl_3_): δ 162.2, 144.5, 129.7, 58.8, 54.0, 46.5, 41.7, 38.5, 33.3, 31.3, 26.4, 25.1, 24.0, 22.9, 19.5, 13.4. LRMS (FAB) *m*/*z* (rel int): (pos) 336 ([C_18_H_34_N_5_O]^+^, 100), 278 ([C_18_H_34_N_5_O − *t*-Bu]^+^, 15). HRMS (FAB) *m*/*z* calcd for C_18_H_34_N_5_O^+^ 336.2758, found 336.2758.

*(S)-4-[4-(4-Isopropyl-2-oxoimidazolidin-1-yl)butyl]-1-butyl-3-methyl-1H-1,2,3-triazolium iodide (***12c***).***12c** was synthesized according to the procedure described for the synthesis of **12a**. **10c** (1.78 g, 5.78 mmol) and MeI (360 µL, 5.78 mmol) were used. Reaction time: 24 h. Yield: 2.39 g, 92%. Colorless oil. TLC: R*_f_* 0.26 (10:1 CH_2_Cl_2_/MeOH). ${\left[\alpha \right]}_{D}^{21.8}$ = −6.0 (c 0.50, MeOH). ^1^H NMR (400 MHz, CDCl_3_): δ 9.39 (s, 1H), 4.69 (t, *J* = 7.6 Hz, 2H), 4.52 (br s, 1H), 4.27 (s, 3H), 3.56 (t, *J* = 8.8 Hz, 1H), 3.46 (qm, *J* = 8.8 Hz, 1H), 3.36 (dt, *J* = 14.4, 7.2 Hz, 1H), 3.19 (dt, *J* = 14.4, 6.0 Hz, 1H), 3.11 (dd, *J* = 8.8, 7.6 Hz, 1H), 3.04 (m, 2H), 2.04 (quintet, *J* = 7.6 Hz, 2H), 1.86 (m, 2H), 1.73 (m, 1H), 1.68 (sextet, *J* = 7.6 Hz, 2H), 1.44 (sextet, *J* = 7.6 Hz, 2H), 0.99 (t, *J* = 7.6 Hz, 3H), 0.93 (d, *J* = 6.8 Hz, 3H), 0.89 (d, *J* = 6.8 Hz, 3H). ^13^C NMR (100 MHz, CDCl_3_): δ 162.5, 144.6, 129.8, 55.9, 54.0, 48.9, 41.8, 38.6, 33.1, 31.3, 26.4, 24.0, 23.1, 19.5, 18.5, 17.9, 13.4. LRMS (FAB) *m*/*z* (rel int): (pos) 322 ([C_18_H_34_N_5_O]^+^, 100), 278 ([C_18_H_34_N_5_O − *i-*Pr]^+^, 10). HRMS (FAB) *m*/*z* calcd for C_18_H_34_N_5_O^+^ 322.2601, found 322.2603.

*(S)-4-[4-(4-Benzyl-2-oxoimidazolidin-1-yl)butyl]-1-butyl-3-methyl-1H-1,2,3-triazolium bis(trifluoromethanesulfonyl)imide (***1a***).* In a 25 mL roundbottom flask, **12a** (463 mg, 930 μmol) and LiNTf_2_ (267 mg, 930 μmol) were dissolved in deionized water (1.6 mL). The reaction mixture was stirred at 40 °C for 24 h. Upon completion of the reaction, the reaction mixture was cooled to rt, and extracted with CH_2_Cl_2_ (5 × 9 mL). The combined organic extracts were dried over anhydrous Na_2_SO_4_, and filtered. The concentration by rotary evaporation afforded a desired **1a** (577 mg, 95%) as a light yellow liquid. TLC: R*_f_* 0.10 (10:1 CH_2_Cl_2_/MeOH). ${\left[\alpha \right]}_{D}^{22.4}$ = −6.5 (c 1.0, CHCl_3_). ^1^H NMR (400 MHz, CDCl_3_): δ 8.40 (s, 1H), 7.31 (m, 2H), 7.26–7.17 (m, 3H), 4.53 (t, *J* = 7.6 Hz, 2H), 4.34 (br s, 1H), 4.21 (s, 3H), 3.94 (m, 1H), 3.57 (t, *J* = 8.8 Hz, 1H), 3.34 (dt, *J* = 14.4, 6.4 Hz, 1H), 3.18 (dt, *J* = 14.4, 6.0 Hz, 1H), 3.17 (dd, *J* = 8.8, 6.4 Hz, 1H), 2.93 (m, 2H), 2.87 (dd, *J* = 13.6, 5.6 Hz, 1H), 2.77 (dd, *J* = 13.6, 8.0 Hz, 1H), 1.99 (quintet, *J* = 7.6 Hz, 2H), 1.77 (m, 2H), 1.69 (m, 2H), 1.41 (sextet, *J* = 7.6 Hz, 2H), 0.99 (t, *J* = 7.6 Hz, 3H). ^13^C NMR (100 MHz, acetone-*d*_6_): δ 162.4, 145.9, 138.6, 130.2, 129.4, 129.1, 127.4, 121.1 (q, *J*_C-F_ = 319.6 Hz), 54.4, 51.9, 50.4, 42.6, 42.5, 38.1, 32.0, 27.2, 24.7, 23.1, 20.0, 13.6. ^19^F NMR (376 MHz, CDCl_3_): δ −79.3. LRMS (FAB) *m*/*z* (rel int): (pos) 370 ([C_21_H_32_N_5_O]^+^, 100), 278 ([C_21_H_32_N_5_O − Bn]^+^, 25). HRMS (FAB) *m*/*z* calcd for C_21_H_32_N_5_O^+^ 370.2601, found 370.2602.

*(S)-4-[4-[4-(tert-Butyl)-2-oxoimidazolidin-1-yl]butyl]-1-butyl-3-methyl-1H-1,2,3-triazolium bis(trifluoromethanesulfonyl)imide (***1b***)*. **1b** was synthesized according to the procedure described for the synthesis of **1a**. **12b** (331 mg, 714 mmol) and LiNTf_2_ (205 mg, 714 mmol) were used. Reaction time: 24 h. Yield: 417 mg, 95%. Colorless oil. TLC: R_f_ 0.56 (10:1 CH_2_Cl_2_/MeOH). ${\left[\alpha \right]}_{D}^{22.9}$ = −5.7 (c 1.0, CHCl_3_). ^1^H NMR (400 MHz, CDCl_3_): δ 8.45 (s, 1H), 4.54 (t, *J* = 7.6 Hz, 2H), 4.34 (s, 1H), 4.22 (s, 3H), 3.45–3.41 (m, 2H), 3.34 (dt, *J* = 14.4, 6.4 Hz, 1H), 3.20 (t, *J* = 6.0 Hz, 1H), 3.16 (m, 1H), 2.96 (m, 2H), 1.99 (quintet, *J* = 7.6 Hz, 2H), 1.79 (quintet, *J* = 7.6 Hz, 2H), 1.71 (m, 2H), 1.42 (sextet, *J* = 7.6 Hz, 2H), 0.99 (t, *J* = 7.6 Hz, 3H), 0.87 (s, 9H). ^13^C NMR (100 MHz, CDCl_3_): δ 162.2, 144.8, 128.3, 119.8 (q, *J*_C-F_ = 319.7 Hz), 58.8, 53.9, 46.3, 41.6, 37.4, 33.2, 31.0, 26.3, 24.9, 23.7, 22.3, 19.4, 13.2. ^19^F NMR (376 MHz, CDCl_3_): δ −79.2. LRMS (FAB) *m*/*z* (rel int): (pos) 336 ([C_18_H_34_N_5_O]^+^, 100), 278 ([C_18_H_34_N_5_O − *t*-Bu]^+^, 13). HRMS (FAB) *m*/*z* calcd for C_18_H_34_N_5_O^+^ 336.2758, found 336.2760.

*(S)-4-[4-(4-Isopropyl-2-oxoimidazolidin-1-yl)butyl]-1-butyl-3-methyl-1H-1,2,3-triazolium bis(trifluoromethanesulfonyl)imide (***1c***).***1c** was synthesized according to the procedure described for the synthesis of **1a**. **12c** (1.92 g, 4.27 mmol) and LiNTf_2_ (1.23 g, 4.27 mmol) were used. Reaction time: 24 h. Yield: 2.46 g, 96%. Colorless oil. TLC: R*_f_* 0.39 (10:1 CH_2_Cl_2_/MeOH). ${\left[\alpha \right]}_{D}^{21.7}$ = −3.9 (c 1.0, MeOH). ^1^H NMR (400 MHz, CDCl_3_): δ 8.34 (s, 1H), 4.52 (t, *J* = 7.6 Hz, 2H), 4.46 (s, 1H), 4.21 (s, 3H), 3.52 (t, *J* = 8.8 Hz, 1H), 3.41 (qd, *J* = 8.8, 1.6 Hz, 1H), 3.33 (dt, *J* = 14.4, 6.4 Hz, 1H), 3.17 (dt, *J* = 14.4, 6.4 Hz, 1H), 3.09 (dd, *J* = 8.8, 7.6 Hz, 1H), 2.94 (m, 2H), 1.98 (quintet, *J* = 7.6 Hz, 2H), 1.77 (quintet, *J* = 7.6 Hz, 2H), 1.73–1.64 (m, 3H), 1.41 (sextet, *J* = 7.6 Hz, 2H), 0.99 (t, *J* = 7.6 Hz, 3H), 0.92 (d, *J* = 6.8 Hz, 3H), 0.88 (d, *J* = 6.8 Hz, 3H). ^13^C NMR (100 MHz, CDCl_3_): δ 162.4, 144.8, 128.2, 119.8 (q, *J*_C-F_ = 319.7 Hz), 55.9, 53.9, 48.7, 41.7, 37.4, 33.0, 31.0, 26.3, 23.6, 22.4, 19.4, 18.2, 17.7, 13.2. ^19^F NMR (376 MHz, CDCl_3_): δ −79.3. LRMS (FAB) *m*/*z* (rel int): (pos) 322 ([C_17_H_32_N_5_O]^+^, 100), 278 ([C_1_H_34_N_5_O − *i*-Pr]^+^, 11), 57 ([C_4_H_9_]^+^, 62). HRMS (FAB) *m*/*z* calcd for C_18_H_34_N_5_O^+^ 322.2601, found 322.2607.

*(S)-4-[4-(4-Benzyl-2-oxoimidazolidin-1-yl)butyl]-1,3-dibutyl-1H-1,2,3-triazolium iodide (***16a***).* In a screw-cap vial, a mixture of **1****0a** (354 mg, 995 μmol) and *n*-BuI (115 μL, 995 μmol) in DMF (100 μL) was stirred at 100 °C. After 24 h, the reaction mixture was cooled to rt. The purification by column chromatography on silica gel (30:1 CH_2_Cl_2_/MeOH) afforded a desired triazolium iodide **16a** (467 mg, 92%) as a yellow oil. TLC: R*_f_* 0.36 (20:1 CH_2_Cl_2_/MeOH). ${\left[\alpha \right]}_{D}^{20.0}$ = −37.2 (c 1.2, CHCl_3_). ^1^H NMR (400 MHz, CDCl_3_): δ 9.60 (s, 1H), 7.33–7.30 (m, 2H), 7.26–7.19 (m, 3H), 4.73 (t, *J* = 7.2 Hz, 2H), 4.48 (t, *J* = 7.2 Hz, 2H), 4.31 (br s, 1H), 3.97 (m, 1H), 3.64 (t, *J* = 8.8 Hz, 1H), 3.35 (dt, *J* = 14.4, 6.8 Hz, 1H), 3.22–3.14 (m, 2H), 2.98 (m, 2H), 2.90 (dd, *J* = 13.2, 5.2 Hz, 1H), 2.77 (dd, *J* = 13.2, 8.4 Hz, 1H), 2.04 (quintet, *J* = 7.6 Hz, 2H), 1.95 (m, 2H), 1.86 (quintet, *J* = 7.2 Hz, 2H), 1.74 (m, 2H), 1.42 (sextet, *J* = 7.2 Hz, 4H), 1.01 (t, *J* = 7.2 Hz, 3H), 0.99 (t, *J* = 7.2 Hz, 3H). ^13^C NMR (100 MHz, CDCl_3_): δ 161.8, 143.9, 137.1, 130.1, 129.1, 128.8, 126.9, 54.1, 51.4, 51.2, 50.4, 42.0, 41.6, 31.4, 31.0, 26.4, 24.2, 22.7, 19.6, 19.5, 13.4, 13.3. LRMS (ESI) *m*/*z* (rel int): (pos) 412 ([C24H38N5O]^+^, 100), 394 ([C24H38N5O − H_2_O]^+^, 3). HRMS (ESI) m/z calcd for C24H38N5O^+^ 412.3071, found 412.3071.

*(S)-4-[4-(4-(tert-Butyl)-2-oxoimidazolidin-1-yl)butyl]-1,3-dibutyl-1H-1,2,3-triazolium iodide (***16b***).***16b** was synthesized according to the procedure described for the synthesis of **16a**. **10b** (161 mg, 500 μmol) and *n*-BuI (58.0 μL, 500 μmol) in DMF (50 μL) were used. Reaction time: 38 h. Yield: 229 mg, 91%. Yellow oil. TLC: R*_f_* 0.16 (20:1 CH_2_Cl_2_/MeOH). ${\left[\alpha \right]}_{D}^{20.0}$ = −38.4 (c 0.82, CHCl_3_). ^1^H NMR (400 MHz, CDCl_3_): δ 9.63 (s, 1H), 4.74 (t, *J* = 7.2 Hz, 2H), 4.48 (t, *J* = 7.2 Hz, 2H), 4.34 (br s, 1H), 3.51–3.45 (m, 2H), 3.36 (m, 1H), 3.21–3.14 (m, 2H), 3.00 (td, *J* = 8.0, 5.2 Hz, 2H), 2.04 (quintet, *J* = 7.2 Hz, 2H), 1.95 (quintet, *J* = 7.2 Hz, 2H), 1.87 (quintet, *J* = 7.2 Hz, 2H), 1.74 (sextet, *J* = 7.6 Hz, 2H), 1.43 (sextet, *J* = 7.2 Hz, 2H), 1.42 (sextet, *J* = 7.2 Hz, 2H), 1.01 (t, *J* = 7.2 Hz, 3H), 0.99 (t, *J* = 7.2 Hz, 3H), 0.88 (s, 9H). ^13^C NMR (100 MHz, CDCl_3_): δ 162.1, 143.9, 130.1, 58.8, 54.1, 51.1, 46.5, 41.5, 33.3, 31.4, 31.0, 26.3, 25.1, 24.2, 22.6, 19.6, 19.5, 13.4, 13.3. LRMS (ESI) *m*/*z* (rel int): (pos) 378 ([C21H40N5O]^+^, 100), 361 ([C21H_39_N5O − H_2_O]^+^, 2). HRMS (ESI) m/z calcd for C21H40N5O^+^ 378.3227, found 378.3227.

*(S)-4-[4-(4-Isopropyl-2-oxoimidazolidin-1-yl)butyl]-1,3-dibutyl-1H-1,2,3-triazolium iodide (***16c***).***16c** was synthesized according to the procedure described for the synthesis of **16a**. **10c** (154 mg, 500 μmol) and *n*-BuI (58 μL, 500 μmol) in DMF (50 μL) were used. Reaction time: 34 h. Yield: 190 mg, 77%. Yellow oil. TLC: R*_f_* 0.2 (10:1 CH_2_Cl_2_/MeOH). ${\left[\alpha \right]}_{D}^{20.0}$ = −39.8 (c 0.54, CHCl_3_). ^1^H NMR (400 MHz, CDCl_3_): δ 9.64 (s, 1H), 4.74 (t, *J* = 7.2 Hz, 2H), 4.48 (t, *J* = 7.2 Hz, 2H), 4.35 (br s, 1H), 3.58 (t, *J* = 8.8 Hz, 1H), 3.45 (qm, *J* = 7.2 Hz, 1H), 3.35 (dt, *J* = 14.4, 2.8 Hz, 1H), 3.18 (dt, *J* = 14.4, 6.4 Hz, 1H), 3.11 (dd, *J* = 8.8, 7.6 Hz, 1H), 2.99 (m, 2H), 2.04 (quintet, *J* = 7.6 Hz, 2H), 1.95 (quintet, *J* = 7.2 Hz, 2H), 1.87 (m, 2H), 1.78–1.65 (m, 3H), 1.42 (sextet, *J* = 7.6 Hz, 4H), 1.01 (t, *J* = 7.2 Hz, 3H), 0.99 (t, *J* = 7.2 Hz, 3H), 0.93 (d, *J* = 6.8 Hz, 3H), 0.90 (d, *J* = 6.8 Hz, 3H). ^13^C NMR (100 MHz, CDCl_3_): δ 162.4, 144.1, 130.2, 56.0, 54.2, 51.3, 49.1, 41.7, 33.2, 31.5, 31.1, 26.5, 24.4, 22.8, 19.7, 19.6, 18.5, 18.1, 13.6, 13.5. LRMS (ESI) *m*/*z* (rel int): (pos) 364 ([C20H38N5O]^+^, 100), 387 ([C20H38N5O + Na]^+^, 1). HRMS (ESI) m/z calcd for C20H38N5O^+^ 364.3071, found 364.3062.

*(S)-4-[4-(4-Benzyl-2-oxoimidazolidin-1-yl)butyl]-1,3-dibutyl-1H-1,2,3-triazolium bis(trifluoromethanesulfonyl)imide (***2a***).* In a screw-cap vial, **16a** (164 mg, 321 μmol) and LiNTf_2_ (92.2 mg, 321 μmol) were dissolved in deionized water (0.54 mL). The reaction mixture was stirred at 40 °C for 24 h. Upon completion of the reaction, the reaction mixture was cooled to rt, and extracted with CH_2_Cl_2_ (3 × 3 mL). The combined organic extracts were dried over anhydrous MgSO_4_, and filtered. The concentration by rotary evaporation afforded the desired NTf_2_ salt **2a** (192 mg, 86%) as a yellow oil. TLC: R*_f_* 0.53 (15:1 CH_2_Cl_2_/MeOH). ${\left[\alpha \right]}_{D}^{20.0}$ = −42.1 (c 1.1, CHCl_3_). ^1^H NMR (400 MHz, CDCl_3_): δ 8.43 (s, 1H), 7.33−7.30 (m, 2H), 7.26−7.17 (m, 3H), 4.54 (t, J = 7.6 Hz, 2H), 4.45 (t, *J* = 7.6 Hz, 2H), 3.96 (m, 1H), 3.58 (t, *J* = 8.8 Hz, 1H), 3.34 (dt, *J* = 14.4, 6.8 Hz, 1H), 3.22−3.14 (m, 2H), 2.92 (m, 2H), 2.87 (dd, *J* = 13.6, 5.2 Hz, 1H), 2.77 (dd, *J* = 13.6, 8.4 Hz, 1H), 1.99 (quintet, *J* = 7.6 Hz, 2H), 1.95 (quintet, *J* = 7.6 Hz, 2H), 1.78 (m, 2H), 1.69 (m, 2H), 1.42 (sextet, *J* = 7.6 Hz, 2H), 1.40 (sextet, *J* = 7.6 Hz, 2H), 1.00 (t, *J* = 7.6 Hz, 3H), 0.99 (t, *J* = 7.6 Hz, 3H). ^13^C NMR (100 MHz, CDCl_3_): δ 161.9, 144.3, 137.0, 129.1, 128.8, 128.6, 126.9, 119.8 (q, *J*_C-F_ = 319.7 Hz), 54.0, 51.4, 50.9, 50.2, 41.9, 41.6, 31.1, 30.8, 26.3, 23.9, 22.3, 19.5, 19.4, 13.3, 13.2. ^19^F NMR (376 MHz, CDCl_3_): δ −79.2. LRMS (ESI) *m*/*z* (rel int): (pos) 412 ([C24H38N5O]^+^, 100), 394 ([C24H38N5O − H_2_O]^+^, 10). HRMS (ESI) m/z calcd for C24H38N5O^+^ 412.3071, found 412.3069.

*(S)-4-[4-(4-(tert-Butyl)-2-oxoimidazolidin-1-yl)butyl]-1,3-dibutyl-1H-1,2,3-triazolium bis(trifluoromethanesulfonyl)imide (***2b***).***2b** was synthesized according to the procedure described for the synthesis of **2a**. **16b** (199 mg, 393 μmol) and LiNTf_2_ (113 mg, 393 μmol) were used. Reaction time: 24 h. Yield: 204 mg, 79%. Light yellow oil. TLC: R*_f_* 0.69 (10:1 CH_2_Cl_2_/MeOH (×2)). ${\left[\alpha \right]}_{D}^{20.0}$ = −68.0 (c 0.40, CHCl_3_). ^1^H NMR (400 MHz, CDCl_3_): δ 8.43 (s, 1H), 4.54 (t, *J* = 7.6 Hz, 2H), 4.46 (t, *J* = 7.6 Hz, 2H), 3.52–3.46 (m, 2H), 3.37 (m, 1H), 3.22–3.14 (m, 2H), 2.92 (td, *J* = 8.0, 4.8 Hz, 2H), 1.99 (quintet, *J* = 7.6 Hz, 2H), 1.95 (quintet, *J* = 7.6 Hz, 2H), 1.79 (sextet, *J* = 7.6 Hz, 2H), 1.71 (m, 2H), 1.43 (sextet, *J* = 7.6 Hz, 2H), 1.40 (sextet, *J* = 7.6 Hz, 2H), 1.01 (t, *J* = 7.6 Hz, 3H), 0.99 (t, *J* = 7.6 Hz, 3H), 0.88 (s, 9H). ^13^C NMR (100 MHz, CDCl_3_): δ 162.5, 144.5, 128.8, 120.2 (q, *J*_C-F_ = 318.9 Hz), 59.4, 54.2, 51.1, 46.7, 42.0, 33.4, 31.3, 31.0, 26.5, 25.1, 24.1, 22.6, 19.7, 19.6, 13.5, 13.4. ^19^F NMR (376 MHz, CDCl_3_): δ −79.2. LRMS (ESI) *m*/*z* (rel int): (pos) 378 ([C21H40N5O]^+^, 100), 383 ([C21H40N5O + Na − H_2_O]^+^, 1). HRMS (ESI) m/z calcd for C21H40N5O^+^ 378.3227, found 378.3237.

*(S)-4-[4-(4-Isopropyl-2-oxoimidazolidin-1-yl)butyl]-1,3-dibutyl-1H-1,2,3-triazolium bis(trifluoromethanesulfonyl)imide (***2c***).***2c** was synthesized according to the procedure described for the synthesis of **2a**. **16c** (169 mg, 343 μmol) and LiNTf_2_ (98.6 mg, 343 μmol) were used. Reaction time: 24 h. Yield: 197 mg, 89%. Colorless oil. TLC: R*_f_* 0.26 (10:1 CH_2_Cl_2_/MeOH). ${\left[\alpha \right]}_{D}^{20.0}$ = −29.7 (c 1.0, CHCl_3_). ^1^H NMR (400 MHz, CDCl_3_): δ 8.44 (s, 1H), 4.55 (t, *J* = 7.2 Hz, 2H), 4.45 (t, *J* = 7.2 Hz, 2H), 3.56 (t, *J* = 8.8 Hz, 1H), 3.46 (dt, *J* = 8.8, 7.2 Hz, 1H), 3.37 (dt, *J* = 14.4, 7.2 Hz, 1H), 3.18 (dt, *J* = 14.4, 6.4 Hz, 1H), 3.12 (dd, *J* = 8.8, 7.6 Hz, 1H), 2.92 (m, 2H), 1.99 (quintet, *J* = 7.2 Hz, 2H), 1.95 (quintet, *J* = 7.2 Hz, 2H), 1.80 (m, 2H), 1.73–1.65 (m, 3H), 1.43 (sextet, *J* = 7.2 Hz, 2H), 1.40 (sextet, *J* = 7.2 Hz, 2H), 1.01 (t, *J* = 7.2 Hz, 3H), 0.99 (t, *J* = 7.2 Hz, 3H), 0.92 (d, *J* = 6.8 Hz, 3H), 0.89 (d, *J* = 6.8 Hz, 3H). ^13^C NMR (100 MHz, CDCl3): δ 162.3, 144.3, 128.5, 119.8 (q, *J*_C-F_ = 319.6 Hz), 56.0, 53.9, 50.9, 48.8, 41.7, 33.0, 31.1, 30.8, 26.3, 24.0, 22.3, 19.5, 19.4, 18.2, 17.7, 13.3, 13.2. ^19^F NMR (376 MHz, CDCl_3_): δ −79.2. LRMS (ESI) *m*/*z* (rel int): (pos) 378 ([C20H38N5O]^+^, 100), 346 ([C20H38N5O − H_2_O]^+^, 1). HRMS (ESI) m/z calcd for C20H38N5O^+^ 364.3071, found 364.3066.

### 3.3. Representative Procedure for Ionic Liquid-Supported Asymmetric Benzylation

In a 1 mL roundbottom flask, benzylimidazolidinone ionic liquid 2a (69.3 mg, 100 μmol) was dissolved in anhydrous THF (0.37 mL) at rt, and *n*-butyllithium solution (44 μL, 110 μmol, 2.5 M in hexanes) was added at −78 °C. The reaction mixture was then stirred for 20 min and propionyl chloride (11 µL, 120 μmol) was added dropwise. After stirring at −78 °C for 6 h, the reaction mixture was warmed to rt, quenched with saturated aqueous NH_4_Cl (1 mL), and diluted with CH_2_Cl_2_ (5 mL). The mixture was basified with 10% aqueous NaOH (3 mL). The aqueous layer was separated, and extracted with CH_2_Cl_2_ (2 × 8 mL). The combined organic layers were washed with brine (8 mL), dried over anhydrous Na_2_SO_4_, filtered, and concentrated in vacuo at 60 °C for 14 h. The colorless liquid 17a was used directly in the next step without further purification. In a 1 mL roundbottom flask, imidazolidinone 17a was dissolved in anhydrous THF (0.32 mL) at rt. The reaction mixture was cooled to −78 °C and NaHMDS solution (180 μL, 180 μmol, 1 M in THF) was added. The mixture was then stirred for 1 h, and a solution of 3-chlorobenzyl bromide (26.2 μL, 199 μmol) in THF (0.08 mL) was added dropwise, warmed to 0 °C, and stirred for 7 h. Upon completion of the reaction, the reaction mixture was quenched with saturated aqueous NH_4_Cl (10 mL), warmed to rt, and extracted with EtOAc (3 × 20 mL). The combined organic layers were washed with brine (30 mL) and dried over anhydrous MgSO_4_, filtered, and concentrated by rotary evaporation. The colorless liquid 18a was used directly in the next step without further purification. In a screw cap vial, benzylated **1**8a was dissolved in a mixture of anhydrous 1,4-dioxane (0.30 mL) and 2 N NaOH (0.30 mL) at rt and the reaction mixture was stirred at 80 °C for 3.5 h. Upon completion of the reaction, the reaction mixture was cooled to 0 °C and extracted with CH_2_Cl_2_ (3 × 4 mL). The aqueous layer was acidified with 10% HCl (5 drops) to pH 2 and extracted with EtOAc (3 × 4 mL). The combined CH_2_Cl_2_ layers were dried over anhydrous Na_2_SO_4_, filtered, and concentrated in vacuo to give crude chiral IL **2a**. The combined EtOAc layers were dried over anhydrous Na_2_SO_4_, filtered, and concentrated in vacuo to yield analytically pure acid (−)-**1**5 as a colorless liquid (18.0 mg, 91%). The recovered ionic liquid **2a** (66.7 mg) was dried in vacuo and reuse for the next cycle. (−)-**1**5: TLC: Rf 0.20 (30:1 CH_2_Cl_2_/MeOH). ${\left[\alpha \right]}_{D}^{20.0}$ = −28.5 (c 1.0, acetone). ^1^H NMR (400 MHz, CDCl_3_): δ 7.24–7.18 (m, 3H), 7.07 (dt, *J* = 14.4, 1.6 Hz, 1H), 3.05 (dd, *J* = 13.6, 6.8 Hz, 1H), 2.78 (sextet, *J* = 7.2 Hz, 1H), 2.67 (dd, *J* = 13.6, 7.6 Hz, 2H), 1.20 (d, *J* = 7.2 Hz, 3H). ^13^C NMR (100 MHz, CDCl_3_): δ 181.9, 141.1, 134.2, 129.7, 129.1, 127.2, 126.7, 41.0, 38.9, 16.5. HRMS (ESI) m/z calcd for C10H10ClO2^−^ ([M − H]^−^) 197.0375, found 197.0383.

### 3.4. Synthesis of N-Phenyl (R)-3-(3-chlorophenyl)-2-methylpropionamide ((R)-**S3**) for HPLC Analysis

In a 1 mL roundbottom flask, acid (*R*)-(−)-15 (16.0 mg, 80.5 μmol), DCC (18.3 mg, 88.6 μmol), and DMAP (0.7 mg, 5.6 μmol) were dissolved in anhydrous THF (0.13 mL) at rt. After aniline (8.2 μL, 88.6 μmol) was added, the mixture was stirred for 2 h at rt. Upon completion of the reaction, the reaction mixture was filtered through a plug of Celite. The filtrate was diluted with Et_2_O (5 mL) and washed with 1 N HCl (2 mL). The organic layer was dried over anhydrous Na_2_SO_4_, filtered, and concentrated in vacuo. The purification by column chromatography on silica gel (9:1 hexanes/EtOAc) afforded a desired (*R*)-S3 (9.5 mg, 43%, 94% ee) as a white solid. TLC: Rf 0.16 (9:1 hexanes/EtOAc). ^1^H NMR (400 MHz, CDCl_3_): δ 7.38 (d, *J* = 8.0 Hz, 2H), 7.30 (m, 2H), 7.22−7.19 (m, 3H), 7.10 (m, 2H), 6.85 (br s, 1H), 3.05 (dd, *J* = 13.6, 8.4 Hz, 1H), 2.73 (dd, *J* = 13.6, 7.2 Hz, 1H), 2.56 (sextet, *J* = 6.8 Hz, 1H), 1.29 (d, *J* = 6.8 Hz, 3H). ^13^C NMR (100 MHz, CDCl_3_): δ 173.4, 141.7, 137.5, 134.3, 129.8, 129.0, 128.9, 127.3, 126.7, 124.4, 120.1, 44.7, 40.1, 17.9. HRMS (ESI) m/z calcd for C16H17ClNO^+^ ([M + H]^+^) 274.0993, found 274.1002.

## 4. Conclusions

The ionic liquid-supported imidazolidinone chiral auxiliaries were developed. The ionic liquid-supported chiral imidazolidinone auxiliaries **1a**–**c** and **2a**–**c** were rationally designed and synthesized in 9 steps from commercially available bromobutane. Asymmetric alkylation–cleavage reactions were performed that produced (−)-3-(3-chlorophenyl)-2-methylpropionic acid (**15**) with high yield and high enantioselectivity. In this study, the problems associated with heterogeneous reactions, such as the difficulty with reaction monitoring, moderate stereoselectivity, nonlinear kinetic behavior, diffusion-limited reactivity, unequal distribution of reagents, low reactivity, extended reaction times, and reagent leaching could be resolved. It is believed that the ionic liquid-supported chiral imidazolidinone auxiliaries could also be widely used for other asymmetric reactions.

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
