# Peer review of "Synthesis of 1,2,3-Triazolium Ionic Liquid-Supported Chiral Imidazolidinones and Their Application in Asymmetric Alkylation Reaction"

_molecules, 2019, doi:10.3390/molecules24183349_

Round 1
Reviewer 1 Report
This is an interesting article on the synthesis of 1,2,3-triazolium ionic liquid-supported imidazolidinone chiral auxiliaries. Their applications in asymmetric alkylation-cleavage reactions were also performed and discussed. It's recommended for publication.
Reviewer 2 Report
Comments and Suggestions for Authors
The manuscript entitled "Synthesis of 1,2,3-Triazolium Ionic Liquid-Supported Chiral Imidazolidinones and Their Application in Asymmetric Alkylation Reaction" by Yunkyung Jeong and Jae-Sang Ryu, designed and synthesized ionic liquid-supported imidazolidinone chiral auxiliaries 1a‒c and 2a‒c in 9 steps and their application in asymmetric alkylation reaction and for other asymmetric reactions.
In my opinion, the manuscript is well written and the data is novel
I have the following comments on the manuscript
There are some typographical errors, it should be considered Add the elemental analysis for the new compounds Add the mass fragmentation for the new compounds Add mechanisms for reactions.
Reviewer 3 Report
The article was describes the use of ionic liquids as way to solve many problems of chiral auxillaries in synthesis - notably cleavage and recovery. The use of ionic liquids also has the advantage over solid phase methods which can interfere negatively with the reaction conditions especially a reaction as complex as low temp alkylation etc.
I found the idea interesting, well executed and well written.
My only query is what are the enantiomeric excesses of the reaction under standard conditions (i.e just the auxiliary)? While the ee's are very good for this reagent it would be nice to be able to compare it with the auxiliary without the attached ionic liquid part. It may be that the comparison study is in the references but i couldn't locate it. Either way it would be good to see in the main article - that way the reader could see if the presence of the ionic liquid part lowers the ee or the yield etc.
With that minor point aside I would recommend publishing the article
